# Relation of Pulmonary Diffusing Capacity Decline to HRCT and VQ SPECT/CT Findings at Early Follow-Up after COVID-19: A Prospective Cohort Study (The SECURe Study)

**DOI:** 10.3390/jcm11195687

**Published:** 2022-09-26

**Authors:** Terese L. Katzenstein, Jan Christensen, Thomas Kromann Lund, Anna Kalhauge, Frederikke Rönsholt, Daria Podlekareva, Elisabeth Arndal, Ronan M. G. Berg, Thora Wesenberg Helt, Anne-Mette Lebech, Jann Mortensen

**Affiliations:** 1Department of Infectious Diseases, Copenhagen University Hospital, Rigshospitalet, 2100 Copenhagen, Denmark; 2Department of Occupational and Physiotherapy, Copenhagen University Hospital, Rigshospitalet, 2100 Copenhagen, Denmark; 3Department of Cardiology, Section for Lung Transplantation, Copenhagen University Hospital, Rigshospitalet, 2100 Copenhagen, Denmark; 4Department of Radiology, Copenhagen University Hospital, Rigshospitalet, 2100 Copenhagen, Denmark; 5Department of Otorhinolaryngology, Copenhagen University Hospital, Rigshospitalet, 2100 Copenhagen, Denmark; 6Department of Clinical Physiology and Nuclear Medicine, Copenhagen University Hospital, Rigshospitalet, 2100 Copenhagen, Denmark; 7Department of Biomedical Sciences, Faculty of Health and Medical Sciences, University of Copenhagen, 2200 Copenhagen, Denmark; 8Centre for Physical Activity Research, Copenhagen University Hospital, Rigshospitalet, 2100 Copenhagen, Denmark; 9Department of Medical Sciences, University of Copenhagen, 2200 Copenhagen, Denmark; 10Department of Medicine, The National Hospital, 100 Torshavn, Faroe Islands; 11Department of Clinical Medicine, Faculty of Health and Medical Sciences, University of Copenhagen, 2200 Copenhagen, Denmark

**Keywords:** SARS-CoV-2, COVID-19, long COVID, SPECT, HR-CT scan, lung function test

## Abstract

A large proportion of patients exhibit persistently reduced pulmonary diffusion capacity after COVID-19. It is unknown whether this is due to a post-COVID restrictive lung disease and/or pulmonary vascular disease. The aim of the current study was to investigate the association between initial COVID-19 severity and haemoglobin-corrected diffusion capacity to carbon monoxide (DLco) reduction at follow-up. Furthermore, to analyse if DLco reduction could be linked to pulmonary fibrosis (PF) and/or thromboembolic disease within the first months after the illness, a total of 67 patients diagnosed with COVID-19 from March to December 2020 were included across three severity groups: 12 not admitted to hospital (Group I), 40 admitted to hospital without intensive care unit (ICU) admission (Group II), and 15 admitted to hospital with ICU admission (Group III). At first follow-up, 5 months post SARS-CoV-2 positive testing/4 months after discharge, lung function testing, including DLco, high-resolution CT chest scan (HRCT) and ventilation-perfusion (VQ) single photon emission computed tomography (SPECT)/CT were conducted. DLco was reduced in 42% of the patients; the prevalence and extent depended on the clinical severity group and was typically observed as part of a restrictive pattern with reduced total lung capacity. Reduced DLco was associated with the extent of ground-glass opacification and signs of PF on HRCT, but not with mismatched perfusion defects on VQ SPECT/CT. The severity-dependent decline in DLco observed early after COVID-19 appears to be caused by restrictive and not pulmonary vascular disease.

## 1. Introduction

After the first wave of the global coronavirus disease 2019 (COVID-19) pandemic, it became increasingly clear that the pulmonary sequelae often persist far beyond the severe acute respiratory syndrome coronavirus 2 (SARS-CoV-2) infection. Apart from the diverse cluster of symptoms collectively coined “long COVID” [1] (breathlessness, chest pain, and fatigue), several studies have documented various degrees of reduced pulmonary diffusing capacity of carbon monoxide (DLco) in previously hospitalised patients up to 12-months post-discharge [2,3,4,5,6,7,8,9]. In many cases, concomitant residual radiological abnormalities are present on high-resolution chest CT, (HRCT) most typically ground-glass opacities (GGO), interlobular septal thickening, and reticulations [4,8].

The mechanisms of post-COVID-19 DLco reduction and the associated symptoms are currently unknown. While previous studies have reported relatively few patients with signs of overt pulmonary fibrosis (PF) on HRCT post-COVID-19 [10], it is still not known if changes on HRCT such as GGO, interlobular septal thickening, and reticulations will remain and for how long. Given that both in situ pulmonary thrombosis and thromboembolism, triggered by aberrations in the coagulation system and pulmonary endothelialitis [11], are considered cardinal in the conspicuous and “silent” hypoxaemia often observed in COVID-19 [12], this may also contribute to late stage changes in lung function. Thus, apart from post-viral PF, persistent pulmonary thromboembolic disease may contribute to persistent DLco reduction and associated symptoms after COVID-19 [6,13].

This paper is the first report from the Danish SECURe (Sequelae of COVID-19, Copenhagen University Hospital, Rigshospitalet) to present a prospective cohort study monitoring the severity and duration of post-COVID complications by the use of extensive clinical, physiological, and radiologic assessments, both in previously hospitalised and non-hospitalised COVID-19 patients.

The aim of the current study was to investigate the association between initial COVID-19 severity and haemoglobin-corrected diffusion capacity to carbon monoxide (DLco) reduction at follow-up. Furthermore, the aim was also to analyse if DLco reduction could be linked to pulmonary fibrosis (PF) and/or thromboembolic disease within the first months after the illness.

## 2. Materials and Methods

### 2.1. Study Design and Setting

The SECURe study is an ongoing prospective cohort study of individuals with polymerase chain reaction (PCR) confirmed SARS-CoV-2 infection conducted at Copenhagen University Hospital, Rigshospitalet, a tertiary health care centre, aimed to assess long-term sequalae of COVID-19.

The protocol was developed based on early reports from China [14,15] and on follow-up data from the first SARS outbreak in Hong Kong in 2002–2003 [16]. In Denmark, as elsewhere, the COVID-19 treatment strategies have been modified during the study period along with the availability of scientific data. Thus, steroids were first implemented from June 2020 [17,18]. Likewise, some patients admitted during the early epidemic were included in the remdesivir trial, the usage of which increased from May 2020 and became widely available from August 2020 [17,19].

Inclusion was closed ultimo March 2021 due to the significant decline in SARS-CoV-2 transmission rates in Denmark and closure of our dedicated COVID-19 ward. We enrolled 190 participants.

### 2.2. Study Participants

All COVID-19 patients admitted to Rigshospitalet, March 2020–March 2021 were invited to participate. Additionally, non-hospitalised SARS-CoV-2 infected patients were offered inclusion with the aim of including 200 patients, ≥2/3 hereof being hospitalised.

Exclusion criteria included dementia, living at an old age facility and being unable to come for follow-up visits.

The initial SECURe study visit was planned to be conducted 3–4 months after SARS-CoV-2 positive testing/post-discharge for non-hospitalised and hospitalised study participants, respectively. Due to a high workload at the participating departments, it was not always possible to adhere fully to this time-plan (see below).

Here, we report on all participants (n = 67) who had completed their first follow-up by 31 December 2020.

### 2.3. Recruitment

Patients were invited to participate in the study at discharge and/or at a post-discharge telephone consultation. Non-admitted patients were identified through the affiliated testing site and by word of mouth among health care personnel.

### 2.4. Data Sources

Age, sex, Charlson co-morbidity index [20], date of testing SARS-CoV-2 positive, initial COVID-19 symptoms and duration thereof prior to admission, treatment during hospitalisation including maximal oxygen demand, ICU admission, mechanical ventilation and/or extra-corporal membrane oxygenation (ECMO) and duration thereof, as well as total duration of hospitalisation were extracted from the participant’s electronic health record. Even though there is now consensus regarding a more advanced disease severity classification system [21,22], this had not yet been established at the time of this study, and we therefore pragmatically used a trinary system to classify the patients according to the clinical severity of the initial COVID-19 disease, similar to previous studies patients not requiring hospitalization (Group I), patients requiring hospitalization but not ICU admission (Group II), and patients requiring both hospitalisation and ICU admission (Group III) [23,24,25,26,27,28,29,30].

At the follow-up visit, participants were questioned about post-COVID-19 symptoms and respiratory complaints according to the chronic obstructive pulmonary disease assessment test (CAT) [31]. Furthermore, participants completed the health-related quality of life SF-36 questionnaire [32], had an extended assessment of physical performance including Hand Grip strength (HGS) and 30-s Sit-To-Stand Test (STS) muscle strength tests and the Six-Minutes’ Walk Test (6MWT) [33,34,35] (Appendix A), lung function testing [36,37,38], HRCT with subsequent scoring [39] and ventilation-perfusion (VQ) scintigraphy [40,41] (Appendix A, and described briefly below).

Participants with signs of post-COVID-19 sequelae were offered re-assessment at 12 months.

### 2.5. Lung Function Testing

Dynamic spirometry, body plethysmography and single breath measurement of DLco were performed in accordance with the ERS/ATS guidelines [36,37,38]. Forced expiratory volume in the first second (FEV1), forced expiratory volume (FVC), FEV1/FVC-ratio, total lung capacity (TLC), residual volume (RV), RV/TLC-ratio, Hb corrected DLco and diffusion coefficient for CO (Kco) were measured. A FEV1/FVC-ratio and a TLC below the lower limit of normal was classified as an obstructive and restrictive ventilation defect, respectively [42,43].

### 2.6. HRCT Chest Scan

HRCT was obtained both after a breath-hold at deep inspiration and deep expiration. The scans were divided into six zones (three on each side), and evaluated for GGO, PF, and honeycombing (HC). PF was indicated by reticulation, traction and bronchiectasi, in combination or separate. For each of these findings, the extent in every zone was scored from 0 to 4 (Appendix A) [39]. All scans were scored by two experienced readers (AK (radiologist) and TKL (pulmonologist)). The readings were carried out as a multidisciplinary reading with consensus. The two readers were blinded to the clinical and functional data.

At the starting point of the SECURe study, there were no validated CT scoring systems in the context of COVID alterations, so we had to choose a system. The scoring system chosen here was based on the system developed in the “Scleroderma Lung Study” [39]. A proportion of scleroderma patients have lung involvement with both GGO of PF and the scoring system was transferable to this population. There is no consensus regarding which scoring system to use, and various methods have historically been used.

### 2.7. VQ Scintigraphy

VQ scintigraphy was conducted as single photon emission computed tomography (SPECT) with a low dose CT used for attenuation correction. The European Association of Nuclear Medicine interpretation criteria were applied [41]. Perfusion and ventilation defects were visually identified, localised, and classified as mismatched (only defect in perfusion), matched (both perfusion and ventilation defects) or inversely mismatched (only defect in ventilation), and sized as subsegmental or segmental. A matched or inversely mismatched ventilation defect was classified as a ventilatory abnormality, regardless of concomitant HRCT findings, while a mismatched perfusion defect without any concomitant signs of fibrosis in the same area on HRCT, including reticulation with or without GGO, was classified as a vascular abnormality, most likely pulmonary embolism. However, if the HRCT showed signs of fibrosis precisely corresponding to a perfusion defect, it was interpreted as a ventilatory abnormality. Various studies have shown that interstitial lung fibrosis may cause mismatched perfusion defects that may incorrectly be interpreted as pulmonary embolism if not correlated to concomitant CT findings [44,45,46]. All scans were read independently by two experienced pulmonary nuclear medicine specialists (JM & RB) and discrepancies were resolved in consensus. The readers were blinded to the clinical and functional data.

### 2.8. Statistical Analyses

All data were entered into REDCap (10.6.18 ©2021 Vanderbilt University, Nashville, TN, USA). Clinical characteristics, lung function, HRCT, VQ scintigraphy, and physical performance were summarised as percentage (n), mean with standard deviation (SD) for normally distributed variables or median [interquartile range, IQR] for non-normally distributed variables. The differences between clinical severity groups were assessed using Fisher’s exact test for dichotomous and categorical data, Kruskal-Wallis H test for non-normally distributed data, or one-way ANOVA for normally distributed data. If a difference was found, bivariate comparisons with Bonferroni correction for multiple comparisons were made. Wilcoxon rank-sum test was used to assess the difference in groups for time from discharge to follow-up. Fisher’s exact test was used to assess the association between VQ defects and HRCT chest findings of GGO and signs of PF. Univariate linear regression models were used to assess the association between CAT score, VQ defects or HRCT findings with DLco. Multivariable logistic regression models were used to assess the association between VQ defects, HRCT findings or DLco with admission to ICU, age and sex.

Data for physical performance were presented as raw scores and presented as % of age and sex adjusted reference norms.

For all data, a two-sided *p* < 0.05 was considered statistically significant. Statistical analyses were performed using STATA 12 (StataCorp., Stata Statistical Software: College Station, TX, USA: StataCorp LLC).

## 3. Results

Patients were evaluated a median 5 months after testing SARS-CoV-2 positive and 4 months after hospital discharge for those admitted (Table 1). Patients from a higher clinical severity group were older, predominantly of male sex, and had greater pre-COVID comorbidity compared with patients from a lighter clinical severity group. Most patients (93%) reported persistent complaints and had a reduced physical performance and lower SpO_2_ and approximately 25% of the patients had not resumed work (Appendix A).

For two study participants, smoking status was not available. Among the remaining participants, only one reported being a current smoker. Previous smoking was, however, often reported with a gradient across the clinical severity groups, 18, 38 and 60 % in Groups I, II and III, respectively.

### 3.1. Lung Function

Half of the patients had an abnormal lung function: 25% in Group I, 47% in Group II, and 79% in Group III (*p* = 0.02) (Table 2). FEV1 was normal in (94%) and not significantly different between groups, but FVC, TLC and RV were progressively lower in the clinical severity group. A reduced DLco was the most common abnormality across groups; the frequency and severity depended on the clinical severity group, notably in patients with a concomitantly low TLC (Table 2). In 75% (21/28) of the patients with a low DLco, there were no signs of either a low FEV1/FVC or a low TLC, and this pattern was not associated with clinical severity.

### 3.2. HRCT

Most patients (63%) had GGO and the frequency depended on the clinical severity group, with GGO being present in all patients in Group III, where the extent of GGO was also rated as higher (*p* < 0.001). Likewise, signs of PF were noted in 44%, also dependent of the clinical severity group (*p* < 0.001) and was observed in all Group III patients. None of the patients in Group III had HC or a history of prior lung disease. PF was associated with the presence of GGO score > 25% (*p* < 0.001) (Appendix A). One third of patients had bronchiectasis, the proportion of which was higher in Group III than Group II (Table 3). Examples of HRCT findings are depicted in Figure 1.

### 3.3. VQ SPECT

Most patients (80%) had a some ventilatory abnormality; this was more common in Group III than in Group I. Vascular abnormalities were rare and not related to the clinical severity group. Ninety-five percent of participants had at least one type of VQ defect with a mean of five, with a higher proportion in Group II than Group I; however, there was no distinct relation between clinical severity group and the specific type of VQ defect. Thus, mismatched perfusion defects were identified in almost 2/3 of patients; this was not related to the clinical severity group, neither was it associated with the presence of matched perfusion defects, GGO nor PF on HRCT (Appendix A). Likewise, the presence of matched VQ defects was neither associated with GGO nor PF on chest HRCT. Only 14% had a normal VQ SPECT, the frequency of which was independent of the clinical severity group (Table 4). Examples of VQ SPECT findings are shown in Figure 2.

### 3.4. Factors Associated with Reduced DLco

In univariate linear regression analysis, reduced DLco was associated with a higher CAT score, the extent of GGO and PF on HRCT, as well as the number of matched, but not mismatched defects on VQ SPECT (Table 5). In multivariable logistic regression, Group III allocation predicted both GGO > 25% on HRCT, the presence of PF, and reduced DLco, but not the presence of defects on SPECT (Table 6). Age, but not sex, was also predictive for GGO > 25% and PF.

## 4. Discussion

In this Danish cohort of patients with mild to severe COVID-19 the majority had subjective health complaints 5 months after testing SARS CoV-2 positive, irrespective of disease severity. The most common lung function abnormality was reduced DLco. Indeed, both the frequency and severity of reduced DLco differed between clinical severity groups, as did HRCT findings of GGO and fibrosis, and the number of matched defects on VQ SPECT. In contrast, the frequency and extent of mismatched perfusion defects and other signs or pulmonary vascular disease were neither related to reduced DLco nor to clinical severity group.

DLco has been reported at various follow-up times after COVID-19. As in the present study, a reduced DLco is typically noted as part of a restrictive lung disease pattern with a reduced TLC, while signs of obstructive lung disease with a concomitantly low FEV1/FVC is rare [2,4,7,47,48,49,50]. We found that the prevalence of reduced DLco was 17% in Group I. Previous studies have likewise found that a reduced DLco is common in this group within the first months after COVID-19 and vary markedly from 6 to 43%. In our study, the prevalence of reduced DLco was 40% and 70% in Group II and III, respectively. This is consistent with previous findings from Germany and USA, where reduced DLco was reported in 1/3 of Group II patients and >90% among Group III patients [24,25,26]. In contrast, one study, reported lower prevalence of reduced DLco in Group III compared to Group II patients [23], perhaps reflecting selection bias in the former group due to a high mortality rate in patients admitted to the ICU in this population. Thus, in the current and other studies, indices of severity, such as ICU admission, high-flow nasal cannula oxygen therapy, mechanical ventilation and duration thereof have been found to predict the prevalence and extent of DLco reduction [8,23]. Of note, DLco has been reported to gradually increase with time in most Group II patients, but it remains pathologically low at 12-month follow-up in more than half of the patients with a reduced DLco at 3-month follow-up [8]. While the exact prevalence estimates are difficult to compare between countries, due to the differences in the extend of the COVID-19 epidemic, healthcare capacity, as well as, preventive, diagnostic, and therapeutic strategies including hospital/ICU admission thresholds, it can be inferred that a pathologically reduced DLco is exceedingly common after COVID-19, and the prevalence increases with the acute phase clinical severity.

GGO was the most common finding in HRCT, which agrees well with other studies conducted at various follow-up times within the first year after infection (1–12 months) [4,6,8,9,25]. In accordance with previous studies [23,24], we found a gradient across the severity groups with a GGO prevalence of 8, 66 and 100% in Groups I, II and III, respectively. GGO indicate localised infection, inflammation, or fluid in the interstitial or alveolar space, none of which are mutually exclusive. They occur from the onset of COVID-19, and GGO may reflect residual changes from the acute infection [8,9]. The extent of GGO after COVID-19 has previously been associated with peak HRCT pneumonia scores during hospitalisation, and the GGO scores gradually decrease over the first 12 months. Moreover, in accordance with previous studies [4,6,8,9,25], GGO provide a mechanistic link to reduced DLco. The same pathological changes within the lung parenchyma that cause GGO may thus also adversely affect DLco.

Fibrosis was another key HRCT finding, in most cases in the form of reticulation. This was not observed in Group I, but was present in 37% of Group II patients, and all Group III patients. We identified a broad spectrum from very little to substantial fibrosis, but without HC, which would have indicated end-stage pulmonary fibrosis. At follow-up five months after testing SARS CoV-2 infected (and four months after discharge (for those admitted)), fibrosis was notably seen in Group III patients, while some studies [6,9,23,24,49], but not all [8], have also found fibrosis in Group II patients. Though group III included individuals with asthma and or current/past tobacco usage, none of them were registered in the electronic patient file system with a chronic lung disease diagnosis, nor was this disclosed at the initial encounter due to COVID-19 (data not shown). It is therefore unlikely that the difference in CT-scan findings between the groups was (fully) due to pre-existing signs of fibrosis among the SECURe patients requiring treatment at the ICU unit.

The presence of pulmonary fibrosis was associated with both the presence of GGO and reduced DLco. We speculate that the presence of GGO and pulmonary fibrosis reflect a spectrum of underlying interstitial lung changes that may lead to varying degrees of restrictive lung disease with reduced DLco in a severity-dependent fashion. Accordingly, it is well established that long-standing pulmonary inflammation may facilitate pulmonary fibrosis [51,52], and, recently, several elevated plasma biomarkers of pulmonary fibrosis have been reported in COVID-19 patients across severity groups in a manner that is associated with the concurrent decline in DLco [26]. However, further evaluation of this link is needed.

Though there is an overlap in the CT features found in conjunction with and at follow-up after various viral infections, including influenza- and coronaviruses, differences also exist [53]. Models have been developed to differentiate between COVID-19 vs. Influenza A (H1N1) pneumonia based on clinical and radiologic features [54]. With the availability of effective and easily accessible microbiological tests, the differentiation based on radiological findings, including CT features, is not necessary. However, identification of the various patterns and understanding the reasons behind it might be helpful for evaluating treatment response.

To the best of our knowledge, this is the first study to report on systematic VQ SPECT/CT in the follow-up of COVID-19 patients. We found that 95 % had V/Q defects, which was slightly more prevalent in Group II and III (though also highly prevalent in Group I). Sixty-six percent had mismatched defects, all of which were small subsegmental and 40 % had matched defects, the majority segmental and larger. In addition, reverse ventilatory mismatched defects were very prevalent (75%). The high frequency of ventilatory defects (matched and reverse matched) might have made it difficult to identify possible associations between mismatched defects and DLco (Table 4). It is well-documented that pulmonary vascular disease may complicate COVID-19 in the acute stage and contribute to hypoxaemia and respiratory failure [55,56,57], but it is unknown whether this also contributes to the post-COVID decline in DLco observed in many patients. In the present study, more than 20% showed evidence of vascular disease, notably mismatched perfusion defects. Apart from in situ thrombosis and/or pulmonary embolism, this may also reflect the long-term effects of the remarkable COVID-19-associated loss of pulmonary microvasculature recently reported and is also consistent with fibrosis-like inflammatory processes in the lung parenchyma [58]. However, this was neither related to the clinical severity group nor to DLco. Rather, reduced DLco was associated with the number of matched VQ defects, indicating ventilatory disturbance, although the association with clinical severity groups was less clear than for HRCT. This provides a functional correlate of the structural lung parenchymal changes seen on HRCT associated with reduced DLco.

There are several study limitations, which may limit the generalisability. Firstly, although all patients discharged from Rigshospitalet were invited to participate, several patient groups were not included in the current analysis, including patients with dementia and patients living at old age facilities. These patients have a higher risk of developing severe COVID-19 and possibly, consequently hereof, more marked long-term sequelae. Conversely, patients with symptoms believed to be related to their COVID-19 might be more inclined to participate. Furthermore, many patients chose not to participate in the study. Among the patients without the need for hospitalisation, there was an overrepresentation of health care workers.

Due to the epidemic and the ensuing strain on the health care system, the follow-up exams could not always be performed at 3–4 months post infection/discharge; however, the divergence from this timing was limited.

## 5. Conclusions

In conclusion, the post-COVID-19 lung is prone to exhibit a severity-dependent decline in DLco approximately five months after testing SARS-CoV-2 positive, which is caused by a fibrosis-like restrictive lung disease and not pulmonary vascular disease. While it remains to be determined to which extent these features of the post-COVID-19 lung are reversible, our results underline the need of preventive measures for severe COVID-19 and targeted post-COVID rehabilitation.

## Figures and Tables

**Figure 1 jcm-11-05687-f001:**
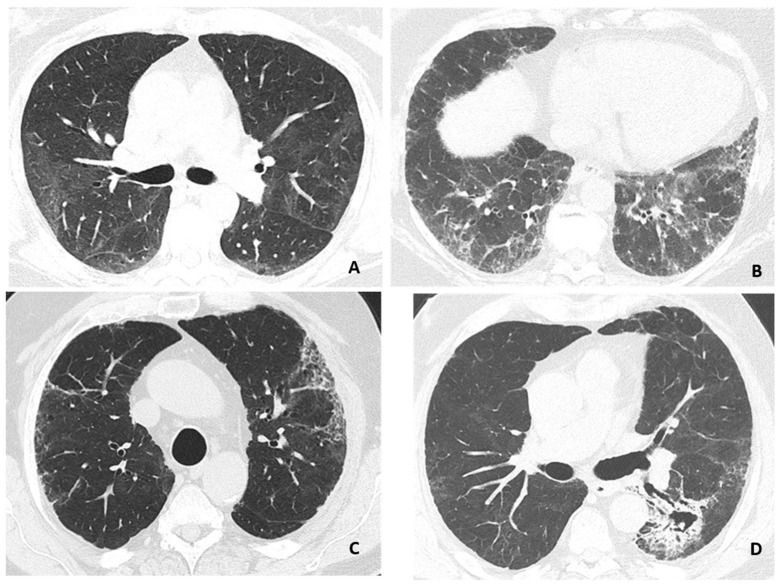
Representative findings on HRCT-scans. (**A**) Ground-glass opacity with discrete interlobular lines. (**B**) Ground-glass opacity with a reticular pattern. (**C**) Discrete ground-glass opacity with reticular pattern and honeycombing. (**D**) Fibrosis with traction bronchiectasis and infarct sequelae with possible fungus ball in the cavity. Images from three patients, (**C**,**D**) is from the same patient.

**Figure 2 jcm-11-05687-f002:**
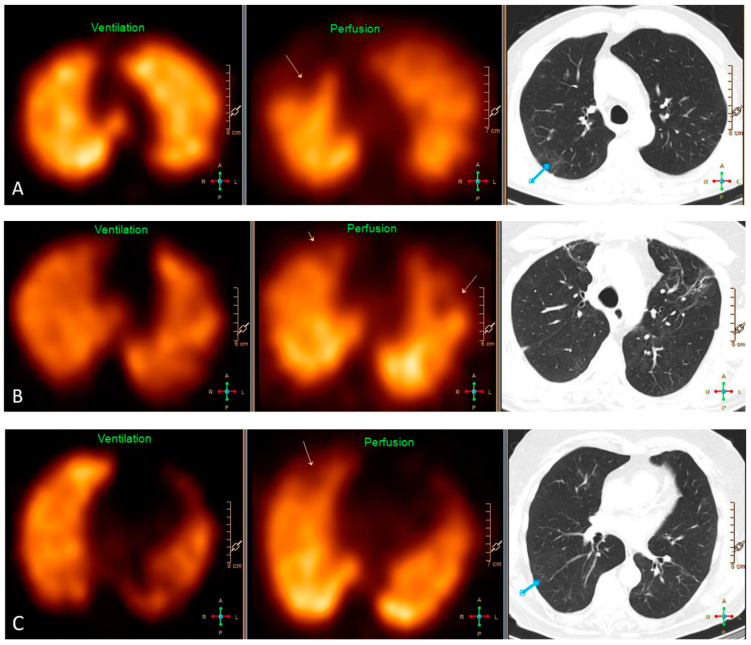
Representative findings on VQ SPECT and HRCT of three patients. (**A**) Pulmonary embolism in the right upper lobe causing a segmental mismatched perfusion defect on SPECT (yellow arrow) without any abnormality in the same area on HRCT. The blue arrow depicts ground-glass opacities dorsally in the right upper lobe without any defect on SPECT. (**B**) HRCT shows signs of fibrosis in the upper lobes causing partially mismatched subsegmental perfusion defects on SPECT (yellow arrows). (**C**) Pulmonary embolism in the right upper lobe causing a subsegmental mismatched perfusion defect on SPECT (yellow arrow) without any abnormality on HRCT in the same area. The blue arrow depicts discrete ground-glass opacities and signs of hypoventilation dorsally in the upper part of the right lower lobe.

**Table 1 jcm-11-05687-t001:** Characteristic of patients with COVID-19 (n = 67) and difference between patients who were not hospitalised, hospitalised without ICU and with ICU treatment.

	All	Group I	Group II	Group III	*p*-Value (between Groups) #
N	67	12	40	15	
Age, years	52.7 ± 14.8	41.8 ± 8.5	54.2 ± 15.6	57.7 ± 12.5	0.012 ^A^
Sex, male	39 (58.2)	3 (25.0)	24(60.0)	12 (80.0)	0.016 ^B^
CCI *†	2 [1;3]	1 [0;2]	2 [0;>3] *	2 [2;>3]	0.073
CAT score *	5 [2;8]	2 [1.5;5.5]	5 [1;6] *	8 [2;10]	0.084
Co-morbidity	36 (53.7)	1 (8.3)	21 (52.5)	14 (93.3)	<0.001 ^C^
Anticoagulation treatment **	29 (46.0)	1 (12.5) **	13 (32.5)	15 (100)	<0.001 ^D^
Before diagnosis **	3 (4.8)	0 (0.0) **	3 (7.5)	0 (0.0)	
After diagnosis **	26 (41.3)	1 (12.5) **	10 (25.0)	15 (100)	
Time from positive SARS CoV-2 PCR test to 3 months follow-up, days	154 [132;191]	175.5 [150;222]	154 [120;187.5]	151 [141;170]	0.349
Time from discharge to follow-up, days ***	130 [98;167]	N/A	139.5 [98;174] ***	113 [95;140]	0.203

Data are expressed as mean ± SD, median [interquartile range] or n (%) as appropriate. CAT score: chronic obstructive pulmonary disease assessment test. † Charlson Comorbidity Index (CCI) values > 3, were recorded as 4 for calculation of the median. * Missing data from one patient (n = 66). ** Missing data from four patients (n = 63). *** Missing data from two patients (n = 53). # Fisher’s exact test, Wilcoxon rank-sum test, Kruskal-Wallis H test or one-way ANOVA where appropriate and if significant followed by bivariate comparison with Bonferroni correction for multiple comparisons. ^A^: Difference between not hospitalised and hospitalised without ICU, and not hospitalised and hospitalised with ICU. ^B^: Difference between not hospitalised and hospitalised with ICU. ^C^: Difference between all groups. ^D^: Difference between not hospitalised and hospitalised with ICU and hospitalised without ICU and hospitalised with ICU.

**Table 2 jcm-11-05687-t002:** Lung function outcome 4 months after COVID-19 (n = 67) and differences between patients who were not hospitalised, hospitalised without ICU and with ICU treatment.

	All (n = 67)	Group I (n = 12)	Group II (n = 40)	Group III (n = 15)	*p*-Value (between Groups) #
FEV1 %P	109.1 ± 19.0	112.5 ± 14.4	109.7 ± 17.5	104.9 ± 25.7	0.564
FVC %P	112.6 ± 20.0	124.8 ± 17.6	112.0 ± 16.5	104.7 ± 26.7	0.031 ^A^
FEV/FVC	79.1 ± 5.7	76.8 ± 5.4	79.3 ± 5.8	80.4 ± 5.3	0.236
TLC %P *	99.9 ± 15.8	113.5 ± 12.2	100.2 ±13.2	87.6 ± 16.4 *	0.001 ^B^
RV %P *	88.5 ± 18.7	99.6 ± 14.6	91.6 ± 17.2	70.4 ± 13.5 *	<0.001 ^C^
RV/TLC %P *	82.9 ± 11.4	84.4 ± 10.5	85.4 ± 11.1	74.7 ± 9.6 *	0.007 ^D^
DLco %P *	79.6 ± 16.7	94.3 ± 16.2	80.3 ± 13.9	64.9 ± 12.6 *	<0.001 ^B^
Kco %P *	92.7 ± 16.2	95.5 ± 17.2	94.1 ± 16.8	86.6 ± 12.7 *	0.272
**Ventilation**					
Restriction *	10 (15.2)	0 (0)	4 (10.0)	6 (42.9) *	0.005 ^C^
Obstruction	2 (3.0)	1 (8.3)	1 (2.5)	0 (0)	0.400
Both restriction and obstruction *	0 (0)	0 (0)	0 (0)	0 (0) *	-
**Diffusion**					
Reduced DLco *	28 (42.4)	2 (16.7)	16 (40.0)	10 (71.4) *	0.014 ^A^
DLco > LLN *	38 (57.6)	10 (83.3)	24 (60.0)	4 (28.6) *	0.020 ^E^
DLco 60%P-LLN *	20 (30.3)	2 (16.7)	13 (32.5)	5 (35.7) *	
DLco < 60 %P *	8 (12.1)	0 (0)	3 (7.5)	5 (35.7) *	
**Both ventilation and diffusion**					
Normal *	33 (50.0)	9 (75.0)	21 (52.5)	3 (21.4) *	0.020 ^A^
Restriction + low DLco *	6 (9.1)	0 (0)	1 (2.5)	5 (35.7) *	0.004 ^C^
Restriction + normal DLco *	4 (6.1)	0 (0)	3 (7.5)	1 (7.1) *	1.000
Obstruction + low DLco *	1 (1.5)	0 (0)	1 (2.5)	0 (0) *	1.000
Obstruction + normal DLco *	1 (1.5)	1 (8.3)	0 (0)	0 (0) *	0.182
Low DLco only *	21 (31.8)	2 (16.7)	14 (35.0)	5 (35.7) *	0.461

Data are expressed as mean ± SD or number (%) when not specified. # Fisher’s exact test or one-way ANOVA where appropriate and if significant followed by bivariate comparison with Bonferroni correction for multiple comparisons. * Missing data from one patient (n = 66). ^A^: Difference between not hospitalised and hospitalised with ICU. ^B^: Difference between all groups. ^C^: Difference between not hospitalised and hospitalised with ICU and hospitalised without ICU and hospitalised with ICU. ^D^: Difference between hospitalised without ICU and hospitalised with ICU. ^E^: Difference between not hospitalised and hospitalised with ICU and between hospitalised without ICU and hospitalised with ICU comparing the normal and moderately-severely reduced DLco.

**Table 3 jcm-11-05687-t003:** HRCT findings in patients 4 months after COVID-19 (n = 63) and differences between patients who were not hospitalised, hospitalised without ICU and with ICU treatment.

	All (n = 64)	Group I (n = 12)	Group II (n = 38)	Group III (n = 14)	*p*-Value (between Groups) #
Any GGO	40 (62.5)	1 (8.3)	25 (65.8)	14 (100)	<0.001 ^A^
Only GGO	12 (18.8)	1 (8.3)	11 (29.0)	0 (0)	0.027 ^D^
>25% GGO *	17 (26.6)	0 (0)	7 (18.4)	10 (71.4)	<0.001 ^B^
Fibrosis (PF + HC)	28 (43.8)	0 (0)	14 (36.8)	14 (100)	<0.001 ^A^
Air trapping	10 (15.6)	2 (16.7)	4 (10.5)	4 (28.6)	0.308
Bronchiectasis	20 (31.3)	3 (25.0)	8 (21.1)	9 (64.3)	0.013 ^C^
Tracheobronchomalacia	5 (7.8)	0 (0)	4 (10.5)	1 (7.1)	0.814
Other **	22 (34.4)	6 (50.0)	11 (29.0)	5 (35.7)	0.441

Data are expressed as n (%). GGO: ground-glass opacities, PF: pulmonary fibrosis, HC: honeycombing. # Fisher’s exact test and if significant followed by bivariate comparison with Bonferroni correction for multiple comparisons. * In more than one zone; ** Noduli, enlarged truncus pulm, emfysem etc. ^A^: Difference between all groups. ^B^: Difference between not hospitalised and hospitalised with ICU and hospitalised without ICU and hospitalised with ICU. ^C^: Difference between hospitalised without ICU and hospitalised with ICU. ^D^: No difference between groups with Bonferroni correction.

**Table 4 jcm-11-05687-t004:** VQ scintigraphy findings in patients 4 months after COVID-19 (n = 65) and differences between patients who were not hospitalised, hospitalised without ICU and with ICU treatment.

	All (n = 65)	Group I (n = 12)	Group II (n = 38)	Group III (n = 15)	*p*-Value (between Groups) #
Ventilatory abnormality	52 (80)	7 (58.3)	30 (79.0)	15 (100)	0.019 ^A^
Vascular abnormality	14 (21.5)	2 (16.7)	11 (29.0)	1 (6.7)	0.215
V/Q defects	62 (95.4)	10 (83.3)	38 (100)	14 (93.3)	0.038 ^B^
*Subsegmental, total*	254	27	153	74	
*Subsegmental, ratio*	4.1	2.7	4.0	5.3	
*Segmental, total*	83	8	51	24	
Mismatched Q defects	43 (66.2)	6 (50.0)	26 (68.4)	11 (73.3)	0.424
*Subsegmental, total*	86	7	53	26	
*Subsegmental, ratio*	2.0	1.2	2.0	2.4	
*Segmental, total*	2	0	2	0	
Matched V/Q defects	26 (40.0)	4 (33.3)	15 (39.5)	7 (46.7)	0.831
*Subsegmental, total*	36	5	18	13	
*Subsegmental, ratio*	1.4	1.3	1.2	1.9	
*Segmental, total*	11	2	3	6	
Reverse mismatched V defects	49 (75.4)	7 (58.3)	30 (79.0)	12 (80.0)	0.353
*Subsegmental, total*	132	15	82	35	
*Subsegmental, ratio*	2.7	2.1	2.7	2.9	
*Segmental, total*	70	6	46	18	
Normal V/Q scan	9 (13.9)	3 (25.0)	6 (15.8)	0 (0)	0.126
Follow-up V/Q scan needed	40 (61.5)	4 (33.3)	24 (63.2)	12 (80.0)	0.050 ^A^

V = ventilation; Q = perfusion. Mismatched Q defects = perfusion defects, but normal ventilation in the area. Reverse mismatched V defects = ventilation defects, but normal perfusion in the area. Data are expressed as n (%), total sum of defects or ratio between number of subsegmental defects and number patient with subsegmental defects. # Fisher’s exact test and if significant followed by bivariate comparison with Bonferroni correction for multiple. ^A^: Difference between not hospitalised and hospitalised with ICU. ^B^: Difference between not hospitalised and hospitalised without ICU.

**Table 5 jcm-11-05687-t005:** Association between CAT score, V/Q scintigraphy defects or HRCT findings with diffusion capacity (DLco %predicted) in patients 4 months after COVID-19 (n = 64) using univariate linear regression.

	B	95% CI	*p*-Value
**Clinical findings**			
CAT score	−0.89	−1.58;−0.19	0.013
**HRCT findings**			
GGO extent *	−1.64	−2.19;−1.10	<0.001
PF extent *	−2.67	−3.74;−1.60	<0.001
**SPECT findings**			
Number of V/Q defects	−1.52	−2.66;−0.39	0.009
Number of mismatched Q defects	−2.09	−5.22;1.03	0.186
Number of matched V/Q defects	−3.69	−7.03;−0.34	0.031
Number of reversed V defects	−0.98	−2.29;0.34	0.143

* Data missing from one patient (n = 63). CAT score: chronic obstructive pulmonary disease assessment test, GGO: ground-glass opacities, PF: pulmonary fibrosis.

**Table 6 jcm-11-05687-t006:** Association between V/Q scintigraphy defects or HRCT findings or diffusion capacity with admission to ICU, age and sex in patients 4 months after COVID-19 (n = 67) using multivariable logistic regression.

	Odds Ratio	95% CI	*p*-Value
Mismatched Q defects **			
ICU admission	1.77	0.47;6.67	0.400
Age in years	1.00	0.96;1.03	0.876
Female sex	1.52	0.50;4.63	0.460
Matched V/Q defects **			
ICU admission	1.64	0.48;5.58	0.427
Age in years	1.00	0.96;1.03	0.845
Female sex	1.49	0.52;4.32	0.459
GGO > 25% ***			
ICU admission	15.48	2.96;80.89	0.001
Age in years	1.10	1.03;1.17	0.003
Female sex	0.59	0.10;3.48	0.561
PF ***			
ICU admission	†	†	†
Age in years	1.10	1.04;1.18	0.002
Female sex	1.96	0.41;9.45	0.402
Reduced DLco *			
ICU admission	4.14	1.07;16.03	0.040
Age in years	1.03	1.00;1.07	0.088
Female sex	0.98	0.32;3.04	0.976

† Omitted from multivariable logistic regression due to collinearity. ICU admission perfectly predicts pulmonary fibrosis (PF). GGO: ground-glass opacities. * Missing data from one patient (n = 66), ** Missing data from two patients (n = 65), *** Missing data from three patients (n = 64).

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
