# Peer review of "Relation of Pulmonary Diffusing Capacity Decline to HRCT and VQ SPECT/CT Findings at Early Follow-Up after COVID-19: A Prospective Cohort Study (The SECURe Study)"

_jcm, 2022, doi:10.3390/jcm11195687_

Round 1

Reviewer 1 Report (New Reviewer)

This study aimed to explore the extent reduction in DLco in COVID19 patients, in related to severity of the disease and the findings of HRCT or SPECT/CT. Combined assessment of HRCT and  SPECT/CT are of value to understand the post COVID19 pathophysiology.

1.The aim is described 'We investigated to which extent the severity of DLco reduction was associated with the presence of persistent 86 symptoms and initial COVID-19 severity, and whether this could be linked to PF and/or pulmonary 87 thromboembolic disease within the first months after the illness.' Are the analysis fully matched to address the aim of this study?

2. The information of smoking history is critical to interpretation of the reduction in DLco.

3. To assess the pathophysiology in decreased DLco, VA is required.

4. Dead space may lead to low Kco, however, mismatched Q defect does not correlate to low DLco. What is the reason for the discrepancy?

5. Is there the value of Spo2 in this study?

6. What are the associated factors with symptoms 5month after follow up? 

7. Low FVC leads to compensation for FEV1/FVC, so that obstruction in airways should be assessed with flow-volume curve but not only FEV1/FVC.

8. Are the findings from SPECT/CT novel? If so, the findings from SPECT/CT would be discussed more in the discussion session. Are any findings from SPECT/CT associated with symptom or clinical outcomes 5 months after ? 

Author Response

This study aimed to explore the extent reduction in DLco in COVID19 patients, in related to severity of the disease and the findings of HRCT or SPECT/CT. Combined assessment of HRCT and SPECT/CT are of value to understand the post COVID19 pathophysiology.

1.The aim is described 'We investigated to which extent the severity of DLco reduction was associated with the presence of persistent 86 symptoms and initial COVID-19 severity, and whether this could be linked to PF and/or pulmonary 87 thromboembolic disease within the first months after the illness.' Are the analysis fully matched to address the aim of this study?

Yes, we have addressed all 4 questions listed in the aims.

DLco reduction was associated with persistent symptoms since linear regression analysis (Table 5) showed that reduced DLco was associated with a higher CAT score at follow-up. DLco reduction was associated to pulmonary fibrosis (on HRCT) but not to thromboembolic disease (mismatched SPECT findings). In addition, we found initial severity of COVID-19 was significantly association with DLco at follow-up, by finding that DLco was more frequently and more severely reduced in Group III (most severely ill) > Group II > Group I (least severely ill).

  1. The information of smoking history is critical to interpretation of the reduction in DLco.

Yes, we agree that there may be an acute adverse effect of smoking on DLco, however, none of the subjects smoked on the day of DLco measurement. We have now added the smoking status of the patients. Only one patient reported ongoing smoking – and did not smoke on the day of testing. However, previous smoking was often reported with a gradient across the three groups.

  1. To assess the pathophysiology in decreased DLco, VA is required.

Yes, to further study the pathophysiological background for a reduced DLCO one can analyse to which degree this is caused by a reduced VA or a reduced KCO or a reduction in both factors. In addition, the reason can be due to reduced DM or capillary blood volume. However, this was not the aim of the present study and due to the relative small sample size, we have chosen not to perform additional sub-analyses of DLCO.

  1. Dead space may lead to low Kco, however, mismatched Q defect does not correlate to low DLco. What is the reason for the discrepancy?

We found relatively few mismatched defects, but many matched and reverse mismatched defects in the patients. The high frequency of ventilatory defects (matched and reverse matched) might have made in difficult to identify possible associations between mismatched defects and DLco (Table 4) given the associations between DLco and matched defects. We have added this to the discussion section.

  1. Is there the value of Spo2 in this study?

These data were already included in the supplementary material, but we have now also included it in the result section.

  1. What are the associated factors with symptoms 5month after follow-up?

Symptoms (CAT score) at 5 months was associated with DLco and CAT score was also associated with initial severity group. 

  1. Low FVC leads to compensation for FEV1/FVC, so that obstruction in airways should be assessed with flow-volume curve but not only FEV1/FVC.

Yes, in some patients with obstructive disease FVC decreases (due to the increase in RV). We did both bodyplethymograpy and flow-volume curves in all subjects so in case patients showed PRISm (preserved ratio impaired spirometry) we could anyhow detect an overt obstructive ventilatory defect from either increased RV/TLC, RV or concave expiratory flow-volume curve with low midexpiratory flow. Yet only two patients were obstructive.

  1. Are the findings from SPECT/CT novel? If so, the findings from SPECT/CT would be discussed more in the discussion session. Are any findings from SPECT/CT associated with symptom or clinical outcomes 5 months after? 

Yes, the findings from the SPECT/CT are novel. We have strengthened the description of this point in the discussion.

Reviewer 2 Report (New Reviewer)

Thank you for the opportunity to review this manuscript. The authors evaluated the association between severity of COVID-19 at diagnosis with the degree of post-COVID respiratory impairment. The study was conducted in a pragmatic/real-world environment, which is both a strength and a limitation. Overall, the manuscript is well written but lacks novelty given the considerable data that are now published on post-COVID lung sequelae. 

1. The use of a pragmatic classification of disease severity at baseline used in this manuscript is understandable. However, can the authors retrospectively reclassify disease severity based on recent consensus (which they have referenced) and present these data?

2. Were the clinical severity groups mutually exclusive? Eg: if a participant started in Group-1 at COVID-19 diagnosis, but was later upgraded to the ICU based on clinical decisions, what severity group did this participant contribute to for the analysis? Please provide these data. 

3. Please provide the inter-quartile range along with median (eg: ling 195)

4. Can the authors comment on whether any individual clinical / epidemiological / risk-exposure variables were independently associated with post-COVID lung sequelae after adjusting for (or stratifying by) baseline disease severity? 

Author Response

Thank you for the opportunity to review this manuscript. The authors evaluated the association between severity of COVID-19 at diagnosis with the degree of post-COVID respiratory impairment. The study was conducted in a pragmatic/real-world environment, which is both a strength and a limitation. Overall, the manuscript is well written but lacks novelty given the considerable data that are now published on post-COVID lung sequelae. 

  1. The use of a pragmatic classification of disease severity at baseline used in this manuscript is understandable. However, can the authors retrospectively reclassify disease severity based on recent consensus (which they have referenced) and present these data?

Since the onset of the COVID-19 pandemic, the classification of COVID-19 severity has been debated and indeed varied between studies, and we welcome formal clinically-based criteria as those proposed by Gandhi et al. [1], which have now been widely adapted. However, when we designed and performed the current study, there was no such consensus, and we pragmatically chose a relatively simple classification, which was used in the majority of contemporary studies and also used by health authorities (including the Danish Health Authorities and the CDC) throughout the world. Apart from the four studies cited in our manuscript as sources of this classification [2–5] this classification is thus directly used in many studies [6–8] , and implicit in others, both original articles, systematic reviews and meta-analyses (apart from those cited in our manuscript, see e.g.: [9–15]). This simple classification permitted us to make comparisons between our present findings and those of other studies in the manuscript. We have thus respectfully chosen to keep the current classification in the revised manuscript, and hope that the editor will find this acceptable. To underline this, we have added this sentence to the manuscript: Even though there is now consensus regarding a more advanced disease severity classification system [16] this had not yet been established at the time of this study, and we therefore pragmatically used a trinary system to classify the patients according to the clinical severity of the initial COVID-19 disease, similar to previous studies [2–5]: patients not requiring hospitalization (Group I), patients requiring hospitalization but not ICU admission (Group II), and patients requiring both hospitalisation and ICU admission (Group III). We have added a reference to the study by Gandhi et al [1] as well as some of the other studies, which have used a similar grading system.

The grading system suggested by Gandhi et al [1] has five severity classes, given our relatively small sample size we have chosen to stick to the planned classification system.

  1. Were the clinical severity groups mutually exclusive? Eg: if a participant started in Group-1 at COVID-19 diagnosis, but was later upgraded to the ICU based on clinical decisions, what severity group did this participant contribute to for the analysis? Please provide these data. 

Yes, the classification is mutually exclusive. The patients were classified according to the worst severity, i.e. if the patient was initially not admitted to hospital and needed treatment in the ICU, the patient was classified as a group III patient.

We have added this information in the manuscript.

  1. Please provide the inter-quartile range along with median (eg: ling 195)

As stated in the method section (under the heading statistical analyses) we have reported either mean with SD or median with IQR.

  1. Can the authors comment on whether any individual clinical / epidemiological / risk-exposure variables were independently associated with post-COVID lung sequelae after adjusting for (or stratifying by) baseline disease severity? 

We have reported age, sex and CAT score at follow-up among study participants. We have now included information about smoking status. Only one patient reported ongoing smoking. However, previous smoking was often reported with a gradient across the three groups.

The study was conducted during the period where the alpha prevailed in Denmark. The Delta variant became the dominant strain from medio 2021 and ultimo 2021 the Omicron subtypes were introduced and quickly became the predominant strain(s) (https://files.ssi.dk/covid19/virusvarianter/status/virusvarianter-covid19-28092021-hk85). It would be of interest to conduct analyses like the ones included in the current study comparing the various strain, but due to the timing of the study relative to the circulating strains, this cannot be done with our data.

References.

  1. Massachusetts, F.; Solomon, C.G.; Gandhi, R.T.; Lynch, J.B.; Del Rio, C. Clinical Practice. n engl j med 2020, 18, 1757–1766, doi:10.1056/NEJMcp2009249.
  2. Chun, H.J.; Coutavas, E.; Pine, A.; Lee, A.I.; Yu, V.; Shallow, M.; Giovacchini, C.X.; Mathews, A.; Stephenson, B.; Que, L.G.; et al. Immuno-Fibrotic Drivers of Impaired Lung Function in Post-COVID-19 Syndrome. medRxiv 2021, 2021.01.31.21250870, doi:10.1101/2021.01.31.21250870.
  3. Labarca, G.; Henríquez-Beltrán, M.; Lastra, | Jaime; Enos, D.; Llerena, F.; Cigarroa, I.; Lamperti, L.; Ormazabal, V.; Carlos Ramirez, |; Espejo, E.; et al. Analysis of Clinical Symptoms, Radiological Changes and Pulmonary Function Data 4 Months after COVID-19. 2021, doi:10.1111/crj.13403.
  4. Morin, L.; Savale, L.; Pham, T.; Colle, R.; Figueiredo, S.; Harrois, A.; Gasnier, M.; Lecoq, A.L.; Meyrignac, O.; Noel, N.; et al. Four-Month Clinical Status of a Cohort of Patients After Hospitalization for COVID-19. JAMA 2021, 325, 1525–1534, doi:10.1001/JAMA.2021.3331.
  5. Munker, D.; Veit, T.; Barton, J.; Mertsch, P.; Mümmler, C.; Osterman, A.; Khatamzas, E.; Barnikel, M.; Hellmuth, J.C.; Münchhoff, M.; et al. Pulmonary Function Impairment of Asymptomatic and Persistently Symptomatic Patients 4 Months after COVID-19 According to Disease Severity. Infection 2022, 50, 157–168, doi:10.1007/S15010-021-01669-8.
  6. Townsend, L.; Dowds, J.; O’Brien, K.; Sheill, G.; Dyer, A.H.; O’Kelly, B.; Hynes, J.P.; Mooney, A.; Dunne, J.; Cheallaigh, C.N.; et al. Persistent Poor Health after COVID-19 Is Not Associated with Respiratory Complications or Initial Disease Severity. Ann. Am. Thorac. Soc. 2021, 18, 997–1003, doi:10.1513/ANNALSATS.202009-1175OC.
  7. Chaudhry, F.; Bulka, H.; Rathnam, A.S.; Said, O.M.; Lin, J.; Lorigan, H.; Bernitsas, E.; Rube, J.; Korzeniewski, S.J.; Memon, A.B.; et al. COVID-19 in Multiple Sclerosis Patients and Risk Factors for Severe Infection. J. Neurol. Sci. 2020, 418, doi:10.1016/J.JNS.2020.117147.
  8. Reilev, M.; Kristensen, K.B.; Pottegård, A.; Lund, L.C.; Hallas, J.; Ernst, M.T.; Christiansen, C.F.; Sørensen, H.T.; Johansen, N.B.; Brun, N.C.; et al. Characteristics and Predictors of Hospitalization and Death in the First 11 122 Cases with a Positive RT-PCR Test for SARS-CoV-2 in Denmark: A Nationwide Cohort. Int. J. Epidemiol. 2020, 49, 1468–1481, doi:10.1093/IJE/DYAA140.
  9. Pérez-González, A.; Araújo-Ameijeiras, A.; Fernández-Villar, A.; Crespo, M.; Poveda, E.; Cabrera, J.J.; del Campo, V.; de Araujo, B.G.; Gómez, C.; Leiro, V.; et al. Long COVID in Hospitalized and Non-Hospitalized Patients in a Large Cohort in Northwest Spain, a Prospective Cohort Study. Sci. Reports 2022 121 2022, 12, 1–8, doi:10.1038/s41598-022-07414-x.
  10. Safont, B.; Tarraso, J.; Rodriguez-Borja, E.; Fernández-Fabrellas, E.; Sancho-Chust, J.N.; Molina, V.; Lopez-Ramirez, C.; Lope-Martinez, A.; Cabanes, L.; Andreu, A.L.; et al. Lung Function, Radiological Findings and Biomarkers of Fibrogenesis in a Cohort of COVID-19 Patients Six Months After Hospital Discharge. Arch. Bronconeumol. 2022, 58, 142–149, doi:10.1016/j.arbres.2021.08.014.
  11. Bergman, J.; Ballin, M.; Nordström, A.; Nordström, P. Risk Factors for COVID-19 Diagnosis, Hospitalization, and Subsequent All-Cause Mortality in Sweden: A Nationwide Study. Eur. J. Epidemiol. 2021, 36, 287–298, doi:10.1007/S10654-021-00732-W/TABLES/4.
  12. Plaçais, L.; Richier, Q.; Noël, N.; Lacombe, K.; Mariette, X.; Hermine, O. Immune Interventions in COVID-19: A Matter of Time? Mucosal Immunol. 2021 152 2021, 15, 198–210, doi:10.1038/s41385-021-00464-w.
  13. Bennett, K.E.; Mullooly, M.; O’Loughlin, M.; Fitzgerald, M.; O’Donnell, J.; O’Connor, L.; Oza, A.; Cuddihy, J. Underlying Conditions and Risk of Hospitalisation, ICU Admission and Mortality among Those with COVID-19 in Ireland: A National Surveillance Study. Lancet Reg. Heal. - Eur. 2021, 5, 100097, doi:10.1016/J.LANEPE.2021.100097/ATTACHMENT/983C6BE4-F33B-4C9B-9B5D-0A122823A3ED/MMC1.DOC.
  14. Abraham, G.R.; Kuc, R.E.; Althage, M.; Greasley, P.J.; Ambery, P.; Maguire, J.J.; Wilkinson, I.B.; Hoole, S.P.; Cheriyan, J.; Davenport, A.P. Endothelin-1 Is Increased in the Plasma of Patients Hospitalised with Covid-19. J. Mol. Cell. Cardiol. 2022, 167, 92–96, doi:10.1016/J.YJMCC.2022.03.007.
  15. Jain, V.; Yuan, J.M. Predictive Symptoms and Comorbidities for Severe COVID-19 and Intensive Care Unit Admission: A Systematic Review and Meta-Analysis. Int. J. Public Health 2020, 65, 533–546, doi:10.1007/S00038-020-01390-7/TABLES/4.
  16. Health Organization, W. Guideline Clinical Management of COVID-19 Patients: Living Guideline, 18 November 2021. 2021.

Round 2

Reviewer 1 Report (New Reviewer)

Most of the comments have been addressed. The consistency in the concept throughout abstract and the last part of the introduction and the conclusions is important. The authors stated in the response to comment 1 that the aim of the study is composed of

four points, which is a little bit confusing. One or two aims which is clearly listed should be described in the last part of the introduction, which should be consistent in the abstract, too. The corresponding conclusions should be described both in the abstract and the last of the discussion session. Please confirm the consistency of the concept or aim of the study and the corresponding conclusions. 

Author Response

Thank you very much for this suggestion.

We have added a sentence about the study aims in both the abstract and the method section.

This manuscript is a resubmission of an earlier submission. The following is a list of the peer review reports and author responses from that submission.

Round 1

Reviewer 1 Report

In this revised manuscript the authors did not take into account several suggestions that can tackle the methodological issues of this study (i.e. VQ SPECT visual scoring and reproducibility of the evaluations) and help to support the conclusions of this manuscript. In addition, COVID severity classification could be easily switched to a more consensual classification as suggested since the authors have all the necessary informations for that (see Data sources section of the manuscript).

Reviewer 2 Report

Katzenstein and colleagues present their findings in a Danish cohort of patients 3-5 months post COVID concluding that patients commonly have DLCO abnormalities which could be attributable to parenchymal abnormalities, rather than from embolic events. Overall, the findings are largely consistent with known published literature and thus represents little novel advancement with their manuscripts. 

I have several major criticisms of this manuscripts:

1. Novelty: as mentioned, the findings of PFT and imaging abnormalities are unfortunately not novel. A very similar study demonstrating essentially the same results can be found here: https://www.tandfonline.com/doi/abs/10.1080/24745332.2022.2054047?src=&journalCode=ucts20

In addition, a recent Radiology publication comparing Xenon MRI and CT imaging in long COVID patients further clarified the diffusion abnormalities relative to parenchymal changes over time https://pubs.rsna.org/doi/full/10.1148/radiol.220069. These findings are also coroborated with an even earlier pre-print: https://www.medrxiv.org/content/10.1101/2022.02.01.22269999v1. Taken all together, there is evidence to suggest diffusion abnormalities on imaging AND PFT can occur in absence of parenchymal disease or restriction. In fact, with the Xenon MRI data, it further raises the question if VQ scans are sensitive enough to detect the microthrombi that have been well documented in post-COVID lung autopsies (reviewed here: https://www.ncbi.nlm.nih.gov/pmc/articles/PMC8089413/). Thus, the conclusion that DLCO abnormalities is associated with fibrotic changes and not pulmonary embolism may not be valid.

2. Pulmonary fibrosis and honeycombing: From a physiologic perspective, it is very difficult to conceive that an acute infection 3 months prior will result in enough inflammation, followed by fibrosis, followed by architectural distortion (ie: honeycombing) for the fibrotic changes to be attributable to COVID infection itself. Serial imaging in patients with RA-ILD shows development of honeycombing over 5.9 years (https://www.ncbi.nlm.nih.gov/pmc/articles/PMC6993451/#:~:text=First%2C%20in%20terms%20of%20radiological,of%205.9%20years%20%5B15%5D.). Thus, the finding of honeycombing in all 14 patients who required ICU care at 3 months post COVID (Table 3) strongly suggests that ILD was a pre-existing diagnosis. Naturally, those patients who have underlying lung disease would be more likely have more severe acute illness, requiring intensive care. Thus, this demonstrates that the are critical confounders between Groups 1 and 3. Hence, the association of DLCO abnormalities with restriction would not be a surprise if these 14 patients in the ICU cohort had ILD to begin with. I recognize the Charlton was used in an attempt to provide information about pre-existing comorbidities but specific documentation of known pre-existing pulmonary disease is of critical importance. As well, patient recall of pre-COVID CAT score (which, as a side note, is an inappropriate dyspnea score since this is a score validated for COPD when essentially none of the patients in this study had obstructive findings - Table 2) would have been helpful to illustrate this issue (though, of course, this would be limited by recall bias)

3. CT interpretation system: I appreciate the author's comments that they empirically chose a CT interpretation algorithm as none was available at the time of study conception. However, given the large number of CT interpretation systems used since specifically for COVID and long COVID, it is difficult to rationalize why the study team cannot re-analyse the images based on the current standards of practice. Furthermore, the adoption of an ILD specific CT interpretation algorithm is flawed given as the authors also admit that to date, there is no clear evidence that an entity of "post-COVID ILD" exists as many have reported improvements in fibrotic changes over time (thus not real "pulmonary fibrosis", which by definition implies irreversibility).